# Prevalence of Frailty and Its Association with Depressive Symptoms among European Older Adults from 17 Countries: A 5-Year Longitudinal Study

**DOI:** 10.3390/ijerph192114055

**Published:** 2022-10-28

**Authors:** Priscila Marconcin, Sharon Barak, Gerson Ferrari, Élvio R. Gouveia, Marcelo de Maio Nascimento, Renata Willig, Margarida Varela, Adilson Marques

**Affiliations:** 1CIPER-Interdisciplinary Centre for the Study of Human Performance Faculty of Human Kinetics, University of Lisbon, 1495-751 Lisbon, Portugal; 2KinesioLab, Research Unit in Human Movement Analysis, Piaget Institute, 2805-059 Almada, Portugal; 3Department of Nursing, School of Health Sciences, Ariel University, Ariel 4076414, Israel; 4Department of Pediatric Rehabilitation, The Edmond and Lily Safra Children’s Hospital, The Chaim Sheba Medical Center, Ramat-Gan 5290002, Israel; 5Escuela de Ciencias de la Actividad Física, el Deporte y la Salud, Universidad de Santiago de Chile (USACH), Santiago 9170022, Chile; 6Facultad de Ciencias de la Salud, Universidad Autónoma de Chile, Providencia 7500912, Chile; 7Department of Physical Education and Sport, University of Madeira, 9020-105 Funchal, Portugal; 8LARSYS-Laboratory for Robotics and Engineering System, Interactive Technologies Institute, 9020-105 Funchal, Portugal; 9Department of Physical Education, Federal University of Vale do São Francisco, Petrolina 56304-917, Brazil; 10RECI—Research Unit in Education and Community Intervention, Piaget Institute, 2805-059 Almada, Portugal; 11ISAMB-Environmental Health Institute, Faculty of Medicine, University of Lisbon, 1649-020 Lisbon, Portugal

**Keywords:** frailty, depressive symptoms, old age, SHARE database

## Abstract

Background: This study aimed to examine the association between frailty and depressive symptoms. Methods: Cross-sectional and five-year longitudinal study. Data were from the population-based Survey of Health, Aging, and Retirement in Europe (SHARE) waves six (2015) and eight (2020). Frailty was assessed using the SHARE-Frailty Instrument. Fatigue, appetite, walking difficulties, and physical activity were self-reported, and grip strength was measured using a handgrip dynamometer. The EURO-D 12-item scale was used to measure depressive symptoms. Results: The sample comprised 25,771 older adults (56.2% female) with a mean age of 66.5 (95% CI 66.4, 66.6) years. The prevalence of frailty was 4.2% (95% CI 3.9, 4.4) in 2015 and 6.7% (95% CI 13.5, 14.3) in 2020. Among frail participants, 72.5% and 69.6% had depression in 2015 and 2020, respectively. Frailty was associated with depression over the 5 years. Those with pre-frailty and those with frailty in 2015 had 1.86 (95% CI 1.71, 2.01) and 2.46 (95% CI 2.14, 2.83) greater odds of having depressive symptoms in 2020. Conclusions: Frailty is a predictor of depression in older adults, and frail participants had greater odds of presenting depressive symptoms.

## 1. Introduction

Older adults face various challenges associated with physical and psychological changes. More specifically, many older adults are at risk of developing frailty and depression—both geriatric syndromes that cause disability and contribute to the global burden of disease [1,2].

Frailty is a biological syndrome characterized by decreased reserve and resistance to stressors across multiple physiologic systems [3]. Frailty is associated with poor health outcomes, such as falls, incident disability, hospitalization, chronic disease, and mortality [4]. The frailty phenotype requires the presence of three or more of the following criteria: unintentional weight loss, self-reported exhaustion, weakness, slow walking speed, and low physical activity [3]. In 2021, the population-level prevalence of frailty using the frailty phenotype measure was 12% (95% confidence interval, CI = 11–13%), while that of pre-frailty was 46% (95% CI = 45–48%) among older adults. The prevalence was higher among females than males [5].

Depression is the most common geriatric psychiatric disorder and a major risk factor for disability and mortality [6]. The prevalence of depression in old age was estimated at 31.74% (95% CI 28–36) [7]. However, the true prevalence of depression in old age is probably higher, as it is usually under-recognized and considered to be a part of the ageing process [8]. Accordingly, depression in older age is commonly undertreated. Pharmacological treatments are efficacious for reducing depressive symptoms [9]. Still, they have many side effects, including falls, fractures, and an increased risk of premature death [10]. Moreover, antidepressant users appeared to be at the highest risk of becoming frail [11].

The overlap of frailty and depression syndromes has been studied, and a positive association between the two has been found [12,13,14]. Therefore, the literature suggests that depression and frailty are interrelated concepts but distinct constructs that are bidirectionally associated [15,16]. Several explanations have been offered for these associations. For instance, depressed patients are more susceptible to frailty as a result of behavioral factors, such as inactivity [17]. Moreover, the two syndromes could share similar symptoms and risk factors, such as fatigue, loss of weight, poor physical activity, and cognitive impairment [18,19]. Furthermore, the two conditions share similar pathophysiological mechanisms, such as inflammation, immunometabolic dysregulation, dopamine depletion, and mitochondrial dysfunction [16,20,21].

Despite advances in understanding how frailty and depression are related, some questions are still unanswered, such as whether frailty and depression are concurrent or if one generally precedes the other. To further understand the two syndromes, longitudinal studies are required. However, such studies are limited, and their findings are controversial. A recent review of six longitudinal studies analyzed frailty as a predictor of depression [16]. Among these studies, five found that frailty was associated with depressive symptoms [22,23,24,25,26], while one did not find this association. One study was carried out with data from Singapore [22] and another with data from China [26]. Four studies had a short follow-up, i.e., less than 2 years [23,25,26,27]. One study was limited to data from the Netherlands [24]. No European study examined the longitudinal association between frailty and depression in a long-term follow-up. Such studies in the European population are also paramount, as cultural background and societal factors play a significant role in the presentation of depression [28] and frailty [29]. Moreover, a better understanding of the relationship between frailty and depression could help us to understand the factors that contribute to the etiology and prognosis of the two syndromes and improve the associated interventions and clinical practice. In this sense, this study aimed to perform a cross-sectional and longitudinal analysis of data from 17 European countries to explore the association between frailty and depressive symptoms.

## 2. Materials and Methods

### 2.1. Data Source and Study Population

The data for the present study were derived from the population-based Survey of Health, Aging, and Retirement in Europe (SHARE) waves six (2015) and eight (2020). SHARE is a multidisciplinary and cross-national panel database of microdata on health, socioeconomic status, and social and family networks among individuals aged 50 years or older. SHARE extracted probability samples from each participating country, allowing inference from the samples to the finite population of Europeans aged 50 years and over. The target population was defined as all individuals born in 1954 or earlier, speaking the official language of the country, and not living abroad or in an institution during the duration of the fieldwork. More details about the project can be found elsewhere [30] and on the website (http://www.share-project.org/home0.html, accessed on 20 October 2022), as well as methodological details for wave 6 [31] and wave 8 [31]. The SHARE protocol was approved by the Ethics Committee of the University of Mannheim and by the Ethics Council of the Max Planck Society for the Advancement of Science, verifying the procedures to guarantee confidentiality and data privacy. Written informed consent was obtained from all participants involved in the study. All procedures were performed following ethical guidelines and regulations in accordance with the Declaration of Helsinki [32]. The first draft of the SHARE questionnaire was piloted with the help of the National Centre for Social Research. Face-to-face interviews to answer the questionnaires, lasting approximately 90 min, were used to collect data at the participants’ homes. Translation experts translated the questionnaires in each country to the respective official language.

Wave six included 68,188 participants, and wave eight included 46,500 participants. The participants were from Austria, Germany, Sweden, Spain, Italy, France, Denmark, Greece, Switzerland, Belgium, Israel, the Czech Republic, Poland, Luxembourg, Slovenia, Estonia, and Croatia. This study’s sample included those who self-reported responses in four of the five frailty constructs (i.e., fatigue, appetite, walking difficulties, and physical activity) and who were physically evaluated for grip strength. In addition, the participants had to report depressive symptoms and information that allowed for their characterization (i.e., sex, age, partner living in the house, categories of body mass index, number of chronic diseases, and self-perceived health). Our study included only samples that participated in 2015 and in 2020. The missing data were not analyzed.

### 2.2. Measures

#### 2.2.1. Frailty

Frailty was assessed by the SHARE-Frailty Instrument (SHARE-FI) [33]. SHARE-FI uses the five constructs from Fried’s frailty phenotype [3]—four self-reported and one physically assessed. Fatigue, appetite, walking difficulties, and physical activity were self-reported. For grip strength evaluation, participants performed two repetitions in each hand using a handgrip dynamometer (Smedley, S Dynamometer, TTM, Tokyo, Japan, 100 kg). A scoring algorithm was used to generate a composite frailty score based on these five assessments, categorizing individuals as non-frail (0 criteria present), pre-frail (1 or 2 criteria present), or frail [33].

#### 2.2.2. Depressive Symptoms

The EURO-D 12-item scale was used to measure depressive symptoms, covering depression, pessimism, suicidality, guilt, sleep, interest, irritability, appetite, fatigue, concentration, enjoyment, and tearfulness (in the last month each item is scored either 0 (‘not present’) or 1 (‘present’). The total score is calculated by summing all of the item scores, ranging from 0 to 12, with a higher score indicating more depressive symptoms. Depressive symptoms were analyzed as a dichotomous variable, and a cutoff of ≥4 points was used to diagnose clinically significant depressive symptoms [34]. Additional details of the scale, as well as its psychometric properties, are described elsewhere [34].

#### 2.2.3. Other Measures

Sociodemographic characteristics: Self-identified sex, age, and living arrangement (living/not living with a partner) were self-reported by the participants.

Height and weight: Participants self-reported their height and weight, and their body mass index (BMI) was calculated (weight in kg/m^2^). BMI was categorized as underweight (<18.5), normal weight (18.5 to 24.9), overweight (25 to 29.9), or obese (≥30) [35].

Chronic diseases: Participants self-reported how many chronic conditions they had. This variable was categorized into less than two and more than two chronic diseases.

Self-perceived health: Participants self-reported their self-perceived health. Responses ranged from 1 (poor) to 5 (excellent). The answers were divided into two groups: one with excellent, very good, and good, and the other with fair and poor.

#### 2.2.4. Statistical Analysis

Descriptive statistics were calculated for the sample on the baseline wave, including means, frequencies, percentages, and 95% CIs for all variables (wave six, 2015). The sample was divided into three groups by level of frailty: non-frail, pre-frail, and frail. Frequencies and percentages in each frailty group were calculated. The normality was tested with the Shapiro–Wilk test. The differences between frailty groups were calculate using a one-way analysis of variance (ANOVA) test with Scheffe’s post hoc test for continuous variables, and the chi-squared (χ^2^) test was performed to compare the distribution of categorical variables. Multiple logistic regression was performed to analyze the cross-sectional association between frailty classification and depressive symptoms. Depressive symptoms were analyzed as a binary dependent variable. The model was adjusted for the following variables: sex, age, partner living in the house, BMI categories, number of chronic diseases, and self-perceived health. Statistical analysis was performed using SPSS version 26. The significance level was set at *p* < 0.05 (2-tailed).

## 3. Results

Table 1 presents the sample characteristics. The final sample comprised 25,771 (56.2% female) older adults aged ≥ 50. The participants were from 17 countries (i.e., Austria, Germany, Sweden, Spain, Italy, France, Denmark, Greece, Switzerland, Belgium, Israel, the Czech Republic, Poland, Luxembourg, Slovenia, Estonia, and Croatia). The mean age of the sample was 66.5 (95% CI 66.4, 66.6) years; 27.4% were in the age group of 50–59 years, 40.4% were aged between 60 and 69 years, 25.7% were aged between 70 and 79 years and 6.6% were aged 80 years and above. Most of the participants (74.5%) lived with a partner in the house. The participants were mainly overweight or obese (65.1%), 45.1% had more than two chronic diseases, and 30.9% evaluated their health as fair or poor.

Table 2 shows the participants’ characteristics according to frailty classification in 2015 and in 2020. The prevalence of frailty was 4.2% (95% CI 3.9, 4.4) in 2015 and 6.7% (95% CI 6.4, 7.0) in 2020. The prevalence of pre-frailty also increased over the 5 years, from 13.9% (95% CI 13.5, 14.3) in 2015 to 16.4% (95% CI 15.9, 16.8) in 2020. Participants who were frail were more likely to be older in 2015 and in 2020 (*p* < 0.001). Moreover, a significant sex difference in the prevalence of frailty (*p* < 0.001) in both 2015 and 2020 was observed. More specifically, frailty was more prevalent in the female sex—84.5% and 85.1% of frail older adults were female in 2015 and in 2020, respectively. Furthermore, significant differences were observed with respect to the depression scale and depression categories in 2015 and 2020 (*p* < 0.001). The mean values of depressive symptoms were higher among frail participants and above the cutoff values for depression (5.0 + 2.3 and 5.0 + 2.5 in 2015 and 2020, respectively). Among frail participants, 72.5% and 69.6% had depression in 2015 and 2020, respectively.

Table 3 presents the odds ratios for the association between frailty and risk of depression in 2015 and for the prospective association between baseline frailty and depressive symptoms in 2020. The results showed that frailty was associated with depression over the 5 years. In a cross-sectional analysis, in 2015, compared to those without frailty, those with pre-frailty and those with frailty had 3.53 (95% CI 3.25, 3.84) and 7.09 (95% CI 6.08, 8.27) greater odds of having depressive symptoms, respectively. In the prospective analysis, those with pre-frailty and those with frailty in 2015 had 1.86 (95% CI 1.71, 2.01) and 2.46 (95% CI 2.14, 2.83) greater odds of having depressive symptoms in 2020, respectively. The analysis by sex showed that men had slightly higher values than women.

## 4. Discussion

This study’s findings suggest that the prevalence of pre-frailty and frailty increased from 2015 to 2020. In both waves, the prevalence of frailty was higher among females. Similarly, a systematic review of longitudinal studies comprising data accumulated from 23 studies found that frailty was more common among females [36]. One explanation for the higher prevalence of frailty among women is that compared with age-matched men, women tend to have poorer health status (e.g., disability) but longer life expectancy (i.e., more resilient). It has been suggested that a combination of behavioral, social, and biological factors might explain this female–male health–survival paradox. Therefore, future studies should focus not only on treatments for frailty but also on sex differences in the effectiveness of such interventions, with a careful focus on sex-specific biopsychosocial concerns [37].

In the present study, the prevalence of frailty in 2015 and 2020 was 4.2% and 6.7%, respectively. Previous studies have reported higher rates of frailty [38,39,40]. Similarly, in a recent systematic review and meta-analysis comprising data from 240 studies reporting 265 prevalence proportions from 62 countries worldwide, representing 1,755,497 participants, the estimated prevalences ranged between 12 and 24% [5]. The incidence of frailty was estimated as 43.4 new cases per 1000 person-years. The differences between studies in the prevalence of frailty may be explained by the different frailty assessments used, as the prevalence of frailty depends—among other things—on the criteria used to calculate it. A large cohort study showed that the difference in the prevalence of frailty between instruments varied from 1.0% using the Study of Osteoporotic Fractures-Frailty Index instrument to 7.0% using SHARE-FI [29]. Using SHARE-FI, we can observe a small increase in the prevalence of frailty from 2020 (our analysis) to 2022 [41]. Our study’s findings corroborate that frailty is increasing and is a serious geriatric health issue. Moreover, incidence rates of frailty also vary depending on countries’ characteristics (e.g., income levels) [42].

The present study found that pre-frail and frail participants had a greater prevalence of depressive symptoms compared with non-frail participants. In addition, the cross-sectional analysis showed that pre-frail and frail participants had greater odds of depressive symptoms than non-frail participants. The prospective analysis (5-year follow-up) showed similar overall results. The analysis by sex indicated that men had a slightly higher odds ratio than women. The association between frailty and depression was also reported in other studies [22,43,44,45]. For example, a Singaporean longitudinal study (4-year follow-up) supports the role of frailty as a predictor of depression. More specifically, in comparison to non-frail individuals, the odds ratios of having depressive symptoms were 1.86 and (95% CI 1.08, 3.20) and 3.09 (CI 1.12, 8.50) among pre-frail and frail individuals, respectively [22]. Similar odds ratios were found in the present study. In a Japanese study with community-dwelling older adults without depressive symptoms, physical frailty was an independent predictor of new incident depressive symptoms [23]. Another study found similar results [24,25,26]. Only one study did not find that frailty predicted depression—an English longitudinal study with 477 community-dwelling older adults. One hypothesis to explain this controversial finding is that they adjusted for 18 potential confounding factors. However, the study indicated that low gait speed was a single physical performance predictor of incident depression [27].

It is crucial to understand why frailty predisposes individuals to depression. First, frail individuals commonly present physical impairment that impacts their social life and connectedness, which both play a pivotal role in the mental health of older adults [46]. Second, several aging-related biomarkers are associated with frailty and depression, such as inflammatory markers, leukocyte telomere length, and lower vitamin D levels [47]. For example, depression and frailty were positively associated with C-reactive protein, interleukin-1, and interleukin-6. The aforementioned associations were bidirectional [48,49]. Other studies have investigated depression as a predictor of frailty and concluded that individuals with depressive symptoms had a higher risk of frailty [11,38,39]. In another study with women aged 70–79 years, depression and anemia increased the risk of frailty, demonstrating that coexisting inflammatory diseases could be risk factors for frailty [43].

Microbiota might also be altered in both frailty and depression. For example, several researchers have suggested that frailty might be caused by chronic inflammation caused by enhanced intestinal permeability resulting from imbalanced gut microbiota. As a result, musculoskeletal system disorders and neurodegenerative diseases might develop. All of the above can predispose individuals to frailty [50,51]. Regarding depression, gut microbiota can affect the nervous system through the gut–brain axis, which consists of the immune system, vagus nerve, and neuroendocrine system, all of which can modify and control cognitive functions such as depression [51,52]. Taken together, both of the presented physiological pathways to frailty and depression (i.e., inflammatory markers and microbiota) indicate a bidirectional association between frailty and depression.

Further emphasizing the importance of identifying depression among frail elderly adults are the results of a recent prospective clinical cohort study that showed that frail, depressed individuals are at increased risk of death compared with non-frail depressed individuals [47]. Other evidence supports this idea, concluding that frailty is a strong predictor of five-year mortality in psychiatric patients [53]. The aforementioned results place frailty as a driver of the risk of death among depressed elderly people. Accordingly, evidence indicates that the risk of mortality decreases considerably after adjusting for frailty [54]. Considering the coexistence of frailty and depression, the common physiological pathways of these two syndromes, and increased mortality among the depressed frail elderly, to improve clinical interventions for frailty, both syndromes should be targeted, and interventions designed to decrease depression should focus on ameliorating frailty rather than treating only the depression. Moreover, frailty interventions should focus on promoting social relationships and improving parameters of physical independence, which may contribute to dealing with depressive symptoms. A recent narrative review highlights the importance of a multidisciplinary treatment for frail patients with depression, including physical exercise and nutritional advice [16].

Some strengths and limitations of this study must be acknowledged. One of the main strengths of this study is its longitudinal design, which made it possible to establish the temporal relationship between frailty and depressive symptoms from an average of five years of follow-up. In addition, this study includes large representative samples from several countries encompassing multiple cultures, guaranteeing good external validity. Therefore, the external validity of this study’s results is a major strength. On the other hand, this study is subject to some limitations. For example, we did not analyze the contribution of each frailty component. Another limitation is that we assessed depressive symptoms using a self-reported questionnaire instead of formal clinical criteria.

## 5. Conclusions

Frail participants had greater odds of presenting depressive symptoms, and frailty was a predictor of depression in older adults. The complex bidirectional association between frailty and depression should be further investigated. Frailty interventions should be alert to depressive symptoms and aim to treat this issue by—among other things—promoting social relationships and improving physical parameters.

## Figures and Tables

**Table 1 ijerph-19-14055-t001:** Total sample characteristics at baseline, 2015 (*n* = 25,771).

Characteristics	*n* (%)
Sex	
Male	11,291 (43.8)
Female	14,480 (56.2)
Age group	66.5 (66.4, 66.6)
50–59 years	7049 (27.4)
60–69 years	10,403 (40.4)
70–79 years	6627 (25.7)
>80 years	1692 (6.6)
Lives with partner	
Yes	19,201 (74.5)
No	6570 (25.5)
BMI categories	
Underweight	230 (0.8)
Normal	8775 (34.1)
Overweight	10,922 (42.5)
Obese	5844 (22.6)
Multimorbidity	
No	14,153 (54.9)
Yes	11,618 (45.1)
Self-perceived health	
Excellent, very good, or good	17,812 (69.1)
Fair or poor	7959 (30.9)

**Table 2 ijerph-19-14055-t002:** Sample characteristics of SHARE participants according to frailty classification in 2015 and in 2020 (*n* = 25,771).

	Wave 6 (2015)				Wave 8 (2020)			
	Not Frail: *n* = 21,107 (81.9%)	Pre-Frail: *n* = 3592 (13.9%)	Frail: *n* = 1072 (4.2%)	*p*-Value	Not Frail: *n* = 19,819 (76.9%)	Pre-Frail: *n* = 4224 (16.4%)	Frail: *n* = 1728 (6.7%)	*p*-Value
Age group, *n* (%)				<0.001 *				<0.001 ^a,^*
50–60 years	6121 (29.0)	773 (21.5)	155 (14.7)		2380 (12.0)	330 (7.8)	81 (4.7)	
61–70 years	8818 (41.8)	1289 (35.9)	296 (27.6)		8094 (40.8)	1221 (28.9)	292 (16.9)	
71–80 years	5084 (24.1)	1144 (31.8)	399 (37.2)		6938 (35.0)	1519 (36.0)	601 (34.8)	
>80 years	1084 (5.1)	386 (10.7)	222 (20.7)		2407 (12.1)	1154 (27.3)	754 (43.6)	
Sex, *n* (%)				<0.001 ^a^				<0.001 ^a^
Male	10,166 (48.2)	959 (26.7)	166 (15.5)		9830 (49.6)	1203 (28.5)	258 (14.9)	
Female	10,941 (51.8)	2633 (73.3)	906 (84.5)		9889 (50.4)	3021 (71.5)	1470 (85.1)	
EURO-D12, mean (SD)	1.76 (1.7)	3.7 (2.3)	5.0 (2.3)	<0.001 ^b^	1.7 (1.6)	3.4 (2.2)	5.0 (2.5)	<0.001 ^b^
Depression, *n* (%)				<0.001 ^a,^*				<0.001 ^a,^*
Yes	3258 (15.4)	1771 (49.3)	777 (72.5)		2945 (14.9)	2281 (54.0)	1203 (69.6)	
No	17,849 (84.6)	1821 (50.7)	295 (27.5)		16,874 (85.1)	1943 (46.0)	525 (30.4)	

Notes: SD, standard deviation. * Scheffe’s post hoc test showed that the differences in the three frailty categories’ comparisons—both for the age and depression variables—in 2015 and in 2020 were statistically significant; ^a^ *p*-value obtained using the chi-squared test; ^b^ *p*-value obtained using one-way ANOVA.

**Table 3 ijerph-19-14055-t003:** Associations between frailty and depressive symptoms using bivariate logistic regression for the total sample and by sex.

	Total Sample				Male				Female			
Frailty Status in 2015	OR (95% CI) for Depression in 2015		OR (95% CI) for Depression in 2020		OR (95% CI) for Depression in 2015		OR (95% CI) for Depression in 2020		OR (95% CI) for Depression in 2015		OR (95% CI) for Depression in 2020	
	Unadjusted Model	Adjusted Model ^a^	Unadjusted Model	Adjusted Model ^a^	Unadjusted Model	Adjusted Model ^a^	Unadjusted Model	Adjusted Model ^a^	Unadjusted Model	Adjusted Model ^a^	Unadjusted Model	Adjusted Model ^a^
Non-frail	1.00 (ref.)	1.0 (ref.)	1.0 (ref.)	1.0 (ref.)	1.00 (ref.)	1.0 (ref.)	1.0 (ref.)	1.0 (ref.)	1.00 (ref.)	1.0 (ref.)	1.0 (ref.)	1.0 (ref.)
Pre-frail	5.32 (4.94, 5.74)	3.53 (3.25, 3.84)	3.05 (2.83, 3.29)	1.86 (1.71, 2.01)	4.53 (4.14, 4.95)	3.76 (3.23, 4.38)	3.21 (2.78, 3.70)	2.04 (1.75, 2.37)	5.84 (5.07, 6.73)	3.49 (3.16, 3.85)	2.64 (2.42, 2.89)	1.81 (1.65, 2.00)
Frail	14.43 (12.55, 16.58)	7.09 (6.08, 8.27)	5.83 (5.14, 6.61)	2.46 (2.14, 2.83)	11.91 (10.20, 13.91)	7.96 (5.62, 11.28)	4.96 (3.64, 6.72)	2.56 (1.85, 3.54)	15.20 (10.94, 21.12)	7.18 (6.03, 8.54)	5.04 (4.39, 5.80)	2.50 (2.14, 2.92)

Notes: CI, confidence interval; OR, odds ratio. ^a^ Adjusted for sex, age, living with a partner, BMI, number of chronic diseases, and self-perceived health.

## Data Availability

http://www.share-project.org/home0.html (accessed on 20 October 2022).

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
