# Peer review of "Prevalence of Frailty and Its Association with Depressive Symptoms among European Older Adults from 17 Countries: A 5-Year Longitudinal Study"

_ijerph, 2022, doi:10.3390/ijerph192114055_

Round 1

Reviewer 1 Report

This study examined the association between frail and depressive symptoms. It is well written and has an adequate sample size.

1. Please indicate any steps taken to address potential sources of bias.

2. Explain how you handled the missing data.

3. Discuss in detail the generalizability (external validity) of your findings.

Author Response

Reviewer 1:

This study examined the association between frail and depressive symptoms. It is well-written and has an adequate sample size.

Answer: Thank you for your revision. 

  1. Please indicate any steps taken to address potential sources of bias.

Answer: We rewrote the material and methods section and added information about the procedures used by the SHARE project. We hope that now it is clearer.

  1. Explain how you handled the missing data.

Answer: The missing data were not analyzed. We add this information on line 119.

  1. Discuss in detail the generalizability (external validity) of your findings.

Answer: Thank you for your comment. The study used a large representative sample from 17 European Countries with 25,771 participants, which had a good external validity to the European old age population. We add this information in the discussion section as a strength of our study.

Reviewer 2 Report

First of all, I would like to thank the authors for the opportunity to review their interesting manuscript "Prevalence of frailty and its association with depressive symptoms among European old adults from 17 countries: a 5-year longitudinal study". The authors in a large sample tried to identify in a prospective study the relationship between frailty and depressive symptoms on average over five years of follow-up. This is a strength of the study, as is the consideration of representative samples from multiple countries across multiple cultures.

However, while reading the manuscript, I had questions and comments to which I would like to receive answers from the authors.

1. The authors poorly described the cohort of examined individuals. References to the SHARE study do not clarify the situation - an article by Börsch-Supan, A., et al. (2013) only describes the first 4 waves of the study. The site (http://www.share-project.org/home0.html ) contains numerous research papers on the included faces on waves 6 and 8, the reader of this manuscript, I think, should not look for this information so confusingly.

2. Based on the article by Börsch-Supan, A., et al. (2013), each wave included patients aged 50 years and older. Thus, different patients were represented in the 2015 cohort and the 2020 cohort. Then a natural question arises - how were the authors able to assess the contribution of initial frailty to the presence of symptoms of depression after 5 years? If the authors evaluated only the cohort that was examined both in 2015 and 2020, then it was necessary to write about it, it is not obvious from the text of the manuscript.

3. Apparently, some of the questions would not have arisen if the authors had presented a flowchart of the patients inclusion in the study.

4. I think that the introduction and discussion do not sufficiently reflect the existing literature on the relationship between frailty and the development of depression. The authors mention only one study in Singapore on this issue (Feng, L., et al., 2014). However, a review has recently been published on this topic (1), which considers not only this study, but also a number of others (2-5). In my opinion, the authors of the manuscript should review these studies and show that this research has contributed something new on this issue.

5. In the summary, the authors conclude that "Frailty interventions should focus on promoting social relationships and improving physical parameters of independence, which may contribute to dealing with depressive symptoms". These are really important thoughts, but they are in no way supported by the data of the authors, since they did not study any frailty interventions. Therefore, these statements are more appropriate in the Discussion section, when considering further research prospects.

Minor:

There is no need to include numbering (2), (3), (4) in the text of the summary, ordinary subheadings are sufficient.

1.      Aprahamian I, Borges MK, Hanssen DJC, Jeuring HW, Oude Voshaar RC. The Frail Depressed Patient: A Narrative Review on Treatment Challenges. Clin Interv Aging. 2022 Jun 22;17:979-990. doi: 10.2147/CIA.S328432.

2.      Makizako H, Shimada H, Doi T, Yoshida D, et al. Physical frailty predicts incident depressive symptoms in elderly people: prospective findings from the Obu Study of Health Promotion for the Elderly. J Am Med Dir Assoc. 2015 Mar;16(3):194-9. doi: 10.1016/j.jamda.2014.08.017.

3.      Borges MK, Romanini CV, Lima NA, et al. Longitudinal Association between Late-Life Depression (LLD) and Frailty: Findings from a Prospective Cohort Study (MiMiCS-FRAIL). J Nutr Health Aging. 2021;25(7):895-902. doi: 10.1007/s12603-021-1639-x.

4.      Collard RM, Comijs HC, Naarding P, et al. Frailty as a predictor of the incidence and course of depressed mood. J Am Med Dir Assoc. 2015 Jun 1;16(6):509-14. doi: 10.1016/j.jamda.2015.01.088.

5.      Chu XF, Zhang N, Shi GP, et al. Frailty and incident depressive symptoms in a Chinese sample: the Rugao Longevity and Ageing Study. Psychogeriatrics. 2020 Sep;20(5):691-698. doi: 10.1111/psyg.12565.

Author Response

Reviewer 2:

First of all, I would like to thank the authors for the opportunity to review their interesting manuscript "Prevalence of frailty and its association with depressive symptoms among European old adults from 17 countries: a 5-year longitudinal study". The authors in a large sample tried to identify in a prospective study the relationship between frailty and depressive symptoms on average over five years of follow-up. This is a strength of the study, as is the consideration of representative samples from multiple countries across multiple cultures. However, while reading the manuscript, I had questions and comments to which I would like to receive answers from the authors.

Answer: Thank you for your very positive evaluation and all your comments. We answered all your questions and changed the manuscript.

  1. The authors poorly described the cohort of examined individuals. References to the SHARE study do not clarify the situation - an article by Börsch-Supan, A., et al. (2013) only describes the first 4 waves of the study. The site (http://www.share-project.org/home0.html ) contains numerous research papers on the included faces on waves 6 and 8, the reader of this manuscript, I think, should not look for this information so confusingly.

Answer: Thank you. We added more information about the cohort. We correct the references with the papers describing the methodology process of waves 6 and 8, that we use in this paper. 

  1. Based on the article by Börsch-Supan, A., et al. (2013), each wave included patients aged 50 years and older. Thus, different patients were represented in the 2015 cohort and the 2020 cohort. Then a natural question arises - how were the authors able to assess the contribution of initial frailty to the presence of symptoms of depression after 5 years? If the authors evaluated only the cohort that was examined both in 2015 and 2020, then it was necessary to write about it, it is not obvious from the text of the manuscript.

Answer: Thank you. You are right. We analyze just the participants that answered in 2015 and in 2020. We added this information in line 119.  

  1. Apparently, some of the questions would not have arisen if the authors had presented a flowchart of the patients inclusion in the study.

Answer: You are right, but we do not have the information to do a flowchart, we just put a filter on the database, and the criteria of this filter are described in lines 113 to 118. 

  1. I think that the introduction and discussion do not sufficiently reflect the existing literature on the relationship between frailty and the development of depression. The authors mention only one study in Singapore on this issue (Feng, L., et al., 2014). However, a review has recently been published on this topic (1), which considers not only this study, but also a number of others (2-5). In my opinion, the authors of the manuscript should review these studies and show that this research has contributed something new on this issue.

Answer: Thank you. We rewrote the introduction and the discussion sections and added those studies. 

  1. In the summary, the authors conclude that "Frailty interventions should focus on promoting social relationships and improving physical parameters of independence, which may contribute to dealing with depressive symptoms". These are really important thoughts, but they are in no way supported by the data of the authors since they did not study any frailty interventions. Therefore, these statements are more appropriate in the Discussion section, when considering further research prospects.

Answer: Thank you. We agree. We removed this statement from the abstract, put it in the discussion section, and added more information to the abstract's conclusion.

Minor:

There is no need to include numbering (2), (3), (4) in the text of the summary, ordinary subheadings are sufficient.

Answer: Thank you. We removed it.

Round 2

Reviewer 2 Report

The authors did a great job of improving the manuscript and answered all my questions. I have no other comments.